# Replicate and Quantize: A Plug-and-Play Strategy for Load Balancing in Sparse Mixture-of-Experts LLMs

## Abstract

While the rapid increase in the number of model parameters poses significant benefits to the development of large language models (LLMs), computational costs are also raised. In order to tackle this difficulty, the sparse mixture-of-experts(SMoE) model was introduced to tackle LLM scaling by activating a subset of experts per input. Therefore, how to leverage the knowledge of multiple experts will be an important topic. Normally, in the most extreme scenario, employing a balanced expert allocation system will result in a time-saving of $n$ times compared to utilizing only a single expert. Thus, in this paper we (1) systematically analyzed the performance and functionality of each expert. (2) Introduced a metric to fill the blank of evaluating load balance for the sparse mixture-of-experts(SMoE) model, based on the observation. (3) Proposed a dynamic plug-and-play strategy that is both trainingless and near-lossless, effectively resolving the load balancing problem, in contrast to previous works that focused on training strategies.

## 1 Introduction

Large-scale language models (LLMs) have become a cornerstone for advancing natural language processing (NLP) tasks ranging from machine translation to mathematical reasoning, owing to their numerous model parameters Wang et al. (2023) Yuan et al. (2023) Imani et al. (2023) Huang & Chang (2022). In face of the computational cost caused by an increasing amount of model parameters, sparse mixture-of-experts (SMoE) architectures Chen et al. (2023) Riquelme et al. (2021) Zhao et al. (2023) have arosed significant attention due to their empirical success in scaling model capacity efficiently. The core idea behind SMoE is its sparse routing strategy, which enables the model to selectively activate a subset of experts (specialized sub-models) for each input. This selective activation mechanism allows SMoE to increase the overall model capacity without a proportional increase in computational cost. As a result, SMoE is considered as an attractive solution for deploying LLMs and becomes widely adopted in the state-of-the-art LLMs Jiang et al. (2024); Dai et al. (2024); Bai et al. (2023).

**Load Imbalance of SMoE.** Despite these advantages, SMoE architectures face a critical challenge: load imbalance among experts Zhou et al. (2022) Fedus et al. (2022), i.e., some experts are overburdened with a disproportionate amount of work while others remain underutilized. This issue affects the inference speed and resource utilization. Therefore, there are some current SMoE model focuses primarily on the routing mechanism's adjustment during the training stage, but it's unpredictable in the inference period Shazeer et al. (2017); Fedus et al. (2022); Lepikhin et al. (2020b); Zhou et al. (2022). Moreover, this imbalance issue will be more clearly demonstrated within a fixed time window in streaming scenarios, which are more common in real-world applications.

**Load Imbalance Score.** Previous works focused only on optimizing the loss of load balance during SMoE model training, with the aim of enhancing the model's overall performance. However, there is no specific metric target to evaluate the routing strategy in the SMoE model. Based on this situation, we propose a metric for evaluating this phenomenon inside the SMoE model.

**Our Observation: Heavy-Hitter vs Important Experts in SMoE.** During the inference period, we can easily observe which experts are activated more than others, a phenomenon we called "heavy-hitter experts." However, are those experts the most important and unique to the task? Does the

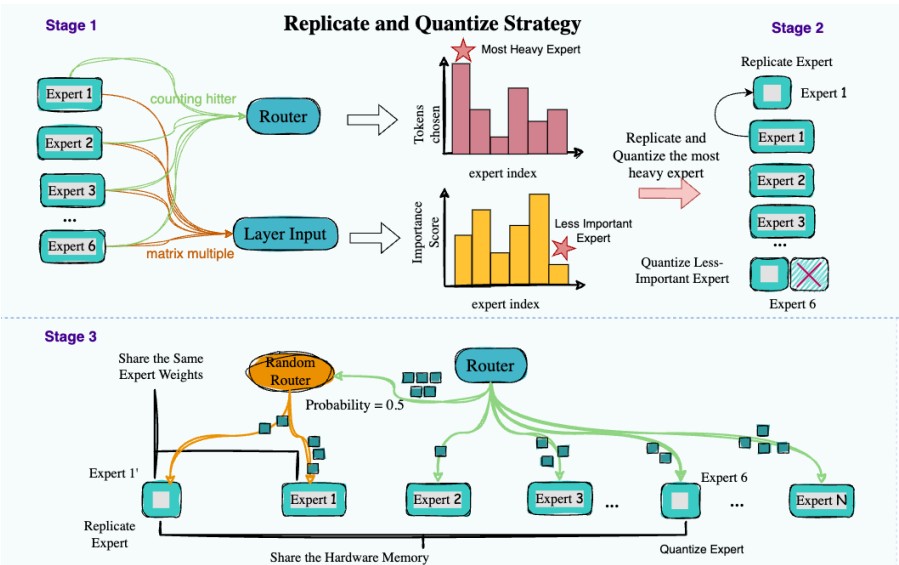

Figure 1: Proposed Replicate and Quanitze pipline.

heavy workload make a significant contribution? In response to these questions, we conducted a series of experiments and observed two axes in the SMoE model—the most important and the heavy hitter. Obviously, the most important experts shoulder the model performance improvement, and the heavy-hitter experts have more workload compared with others.

**We Propose Replicate and Quantize: A Plug-and-Play Strategy for SMoE.** With the uniform metric, we suggest a new plug-and-play strategy that effectively and dynamically solves the load imbalance issues in the SMoE model. Our approach focuses on identifying and optimizing the use of the model's heaviest and least important experts. Specifically, we introduce a low-cost method to pinpoint the most heavily used experts and then replicate these experts using a lower-bit quantized version to mitigate their load. Simultaneously, we quantize the least important experts to ensure that the overall model fits within the total memory budget, thereby maintaining efficiency.

A series of empirical experiments demonstrate that our proposed strategy effectively addresses the load imbalance issue with minimal impact on model performance. By providing a near-lossless solution to redistribute the computational load among experts, our approach enhances the efficiency and practicality of deploying SMoE models. This work contributes to the ongoing efforts to optimize large language models, making them more robust and scalable for real-world applications.

**Summary of Contributions.** We summarize our contributions as below.

- We observe that there are two angles in addressing the load imbalance issue of SMoE: we need to determine the heavy-hitter and important experts in SMoE. Moreover, we perform an analysis on the relationships of these two types of experts in SMoE LLMs.
- We propose a replicate and quantize: a plug-and-play strategy for load balancing in SMoE LLMs. We replicate the heaviest expert with a lower-bit quantized version. Furthermore, we quantized the least important expert to fit the total memory budget. Our empirical evaluation suggests that our approach provides a near-lossless way of addressing the load imbalance issue in SMoE LLMs.
- We demonstrate how the proposed replicate and quantize strategy performs in a streaming setting and guide the SMoE to dynamically manage the load balance in different workloads. Empirical results suggest that our strategy significantly reduce the load imbalance during the workload.
- We provide a metric that effectively evaluates the load balance in the SMoE model, thus filling the gap in evaluating the SMoE model's routing strategy.

## 2 RELATED WORK

**Sparse Mixture-of-Experts (SMoE).** The Mixture-of-Experts model, introduced by Jacobs et al Jacobs et al. (1991), aims to divide the problem into simpler sub-problems. By including sparsity

in the MoE model Shazeer et al. (2017), where only a subset of experts is activated, computing efficiency is significantly enhanced Lepikhin et al. (2020a). Consequently, achieving a comparable balance among the experts becomes a great challenge. Recently, several works have been carried out, to ensure load balance during model inference. For instance, integrate an auxiliary loss function to regulate expert decisions during the model's training Fedus et al. (2022) Lepikhin et al. (2020b) Zhou et al. (2022); limit the expert capacity to prevent a select few experts from being overloaded Lepikhin et al. (2020b); Using random top-k choices, such as top-2 choices, can enhance the probability that an expert will be selected Team (2023) Zhou et al. (2022),or directly relies on stochastic processes instead of deterministic routing to improve the model's generalisation Zuo et al. (2021) Chen et al. (2023) Roller et al. (2021). Meanwhile, SMoE has been used for various applications Zhao et al. (2023), such as image segmentation and object recognition Eigen & Fergus (2015) Riquelme et al. (2021) Zhu et al. (2023), where the ability to focus on specific features of the input is crucial. Meanwhile, this advantage also attracts lots of LLMs' workers to accelerate the model's running time; they can add more parameters to achieve a higher score with less constant running time **?** Dai et al. (2024).

**Improving Efficiency of LLM.**   Enhancing the efficiency of large language models involves optimising hardware and developing algorithmic breakthroughs. In terms of leveraging hardware, using multiple GPUs or TPUs  Jouppi et al. (2017) significantly increased inference speed. Additionally, optimising model architecture can be an effective method. For example, quantization Gong et al. (2014) Jacob et al. (2018) Zhou et al. (2017) Krishnamoorthi (2018), which reduces the precision of numbers used in computations, decreases the size of a model and increases the speed of inference. Additionally, algorithms such as pruning  Han et al. (2015) Molchanov et al. (2016) Frankle & Carbin (2018) and knowledge distillation Hinton et al. (2015) Zagoruyko & Komodakis (2016) Polino et al. (2018) reduce the computational burden by removing irrelevant weights and teaching smaller models to imitate larger ones, respectively, which can still maintain or even improve the model's performance. Sparse training Mocanu et al. (2018) Bellec et al. (2017) Mostafa & Wang (2019) was widely used to decrease the computational cost for the large models, which was performed by reducing the number of active neurons.

# 3   REPLICATE AND QUANTIZE

In this section, we present our plug-and-play strategy for load balancing in sparse mixture-of-experts LLMs. We begin by demonstrating the difficulty of traditional learning-based load balancing strategies in the setting of pre-trained LLMs. Next, we demonstrate how to identify the heavy-hitter experts and the least important experts in SMoE model. Then, we show how an expert's workload and importance shape our two perspectives of view. Finally, we introduce our proposed replicate and quantize algorithm.

## 3.1   LOAD IMBALANCE IN SMOE MODEL

To begin with, we formally define a quantitative metric for load imbalance of SMoE models.

**Definition 3.1** (Load Imbalance Score)**.** *Let $\mathcal{M}$ denote a sparse mixture-of-expert (SMoE) model with $p$ MoE blocks. In each SMoE block, there are $m$ expert network modules. Each input token selects $k < m$ expert in each SMoE block for computation. Given a dataset $X$ with $n$ tokens, we define the number of tokens that select expert $j \in [m]$ at block $i \in [p]$ as $n_{i,j}$. Then we define the load imbalance score for block $i$ as*

$$l_i = \frac{m \max_{j \in [m]} n_{i,j}}{nk}.$$

*Here $\max_{j \in [m]} n_{i,j}$ denote the heaviest expert that receives the most input tokens. Moreover, we know that $nk/m$ represents the ideal case that every expert receives the same number of tokens since $nk$ is the total workload size and $m$ is the number of experts.*

According to the definition, the load imbalance score measures the ratio of the maximum expert workload versus the ideal, averaged expert workload. A higher load imbalance score means poor load balance in the inference phase of SMoE. For a perfectly balanced SMoE, its load imbalance score for every layer should be 1.

Table 1: Fine-Tuning Switch Transformer for Load Balancing

| Method | Load Imbalance Score (See Definition 3.1) | | | | | | | Accuracy |
|--------|-------|-------|------------|--------|---------|-----------|------------|---------|
| | GSM8K | MMLU | Truthful QA | PIQA | Wiki QA | Hellaswag | Winogrande | Wiki QA |
| Tune router 2nd | 2.4556 | 2.8854 | 2.4165 | 2.4814 | 2.7746 | 2.6985 | 2.8346 | 0.2039 |
| Freeze router | 1.8976 | 1.6155 | 1.4866 | 1.5417 | 1.3630 | 1.3925 | 1.4697 | 0.2035 |
| Tune router 10ep | 2.4935 | 2.7618 | 2.3970 | 2.1930 | 2.8080 | 2.4694 | 2.8425 | 0.2133 |
| Tune both | 2.3379 | 3.2713 | 2.8660 | 2.6816 | 3.2152 | 2.8884 | 3.1544 | 0.2062 |
| Freeze router 1ep | 2.1052 | 1.8082 | 1.6714 | 1.5206 | 1.5710 | 1.5718 | 1.7276 | 0.2057 |
| Tune router | 2.7509 | 3.7665 | 3.5322 | 3.3336 | 3.9062 | 3.3334 | 4.1324 | 0.1135 |
| Tune expert | 1.9614 | 1.7298 | 1.6715 | 1.6909 | 1.5011 | 1.6527 | 1.6455 | 0.1955 |
| Full finetune | 2.3056 | 2.7234 | 2.4218 | 2.1323 | 2.6271 | 2.3064 | 2.6285 | 0.2011 |
| Original | 1.9709 | 1.5405 | 1.4956 | 1.5770 | 1.3910 | 1.4182 | 1.5261 | 0.1396 |
| Our method | 1.3937 | 1.2962 | 1.3494 | 1.2756 | 1.2864 | 1.3623 | 1.2146 | 0.1935 |

**The Hardness of Fine-Tuning for Load Balancing.** We show with experiments that it is hard to further fine-tune the SMoE to enforce load balance. We take the Switch Transformer Fedus et al. (2022) as an example and explore a series of fine-tuning strategies based on load-balancing loss proposed in Fedus et al. (2022). As shown in Table 1, we present the results with the following fine-tuning strategies: (1) **Full Finetune:** All the parameters in this model participate in the fine-tune, (2) **Tune Expert:** Only the experts' weights should be tuned. (3) **Tune Router:** Only the router's weights should be tuned. (4) **Freeze Router 1ep:** Only freeze the router weights in the first epoch, and then release them to finish the full fine-tuning. (5) **Tune Both:** Only tune the router and experts' weights. (6) **Tune Router 10ep:** During each epoch, tune the router weights only in the 10% step while freezing the remaining weights. For the remaining steps, finish fine-tuning the entire model. (7) **Freeze Router:** In this fine-tuning strategy, we only freeze the router weights and tune all of the others' weights. (8) **Tune Router 2nd:** Before the last two epochs, we fully fine-tuned the model, and in the last two epochs, we only adjusted the router weights, freezing the other weights.

This table illustrates the performance of the fine-tuned model. Except for the "Tune Router" strategy, the others all have obvious improvements in accuracy, as demonstrated by the results. However, when we used those fine-tuning strategies, almost all of the load imbalance scores (see Definition 3.1) for each dataset in each strategy increased. This suggests that despite having sufficient computing resources and time to refine the model in various ways, achieving the load balance in token allocation remains challenging.

## 3.2 THE HEAVY-HITTER EXPERT ORACLE IN SMoE

We argue that due to the load imbalance of SMoE. There exists a heavy-hitter expert. Formally, we introduce the following oracle.

**Oracle 3.2** (Heavy-Hitter Expert). *Let $\mathcal{M}$ denote a sparse mixture-of-expert (SMoE) model with $p$ MoE blocks. In each SMoE block, there are $m$ expert network modules. Each input token selects $k < m$ expert in each SMoE block for computation. Given a dataset, $X$ with $n$ tokens, the expert $j$ at a layer at MoE block $i$ that has the maximum $n_{i,j}$ among block $i$ is viewed as the heavy-hitter expert in block $i$.*

Next, we describe how to identify heavy-hitters using input data. As shown in Algorithm 1, given an input set of tokens $X$, we keep track of the expert choices for each input $x \in X$. We then quantify an expert's workload by counting the number of input tokens that have selected this expert. For each MoE block, we designate the expert with the heaviest workload as the heavy-hitter for that block. According to our study, we can select up to 10% of the tokens and estimate the heavy-hitter. As a result, we can use Algorithm 1 to retrieve the heavy-hitter expert as suggested in Oracle 3.2.

## 3.3 QUANTIFYING EXPERT IMPORTANCE IN SMoE

We argue that there is another angle we need to consider in load balancing SMoE: the importance of experts. Here we quantify the importance of an expert following a straightforward rule: if we remove an expert in a MoE block and experience a significant accuracy drop in the end-to-end SMoE predictive performance, we say that the removed expert is an important expert. Compared to the approaches mentioned in those training-based strategies, we propose a gradient-free metric that,

modified from the Wanda metric, has shown outstanding effectiveness in extracting less important experts. We compared this method with random-selected experts and determined that the heavy-hitter expert was the less important one, which is shown in the table 2.

---

**Algorithm 1** Search for Heavy-Hitter Expert

---

**Input:**
   $X$ = Input tokens, $En$ = Expert numbers, $L$ = MoE Layers, $T$ = Token numbers, $s$ = Sparsity factor
**Output:** List of Heavy Experts $EC$
**Initialize:** $EC \leftarrow \text{list}[L]$
**for** $l \in L$ **do**
   $expert\_chosen \leftarrow []$
   **for** $x \in X$ **do**
      $expert\_chosen \leftarrow l(x)$
   **end for**
   $expert\_num \leftarrow \text{count}(expert\_chosen)$
   $heavy\_expert \leftarrow \arg\max(expert\_num)$
   $EC[l] \leftarrow heavy\_expert$
**end for**
**return** $EC$

---

Let $W$ be the weight matrix with dimensions $C_{\text{out}} \times C_{\text{in}}$, where $C_{\text{out}}$ is the number of output channels and $C_{\text{in}}$ is the number of input channels. And $X$ be the input matrix with dimensions $(N \cdot L) \times C_{\text{in}}$, where $N$ denote batch size and $L$ the length of each input sequence. The $s$ be the desired sparsity level, a fraction between 0 and 1 indicating the portion of weights to be pruned. We calculate the Wanda metric $S$ via element-wise multiplication between absolute value of $W$ and $\|X\|_2$ which is L2 norm of $X$, computed as:

$$\|X\|_2 = \sqrt{\sum_{i=1}^{N \cdot L} X_{ij}^2} \quad \text{for each column } j.$$

After calculating the metric, the indices of $W$ are sorted in ascending order based on their metric values for each output feature. It then selects the indices with the smallest metric values, which correspond to the weights that have the least impact on the output. With the pruned indices, we extract the corresponding values in the weight matrix $W$. Then, we calculate the mean value of those pruned weights. In this setting, a larger mean value indicates that the expert contains more redundant information. And the expert with the largest mean value can be determined to be the less important expert. We summarize the whole procedure in Algorithm 2.

---

**Algorithm 2** Search for Less Important Expert

---

**Input:** Layer inputs $X$, All experts' weights $W$, MoE Layer $L$, Experts number $E$
**Output:** List of Less Important Expert $IE$
**Initialize:** $IE \leftarrow \text{list}[L]$, $S \leftarrow \text{list}[L][E]$, $IS \leftarrow \text{list}[L][E]$
Extract the Less Important Expert $e_1, e_2, .., e_n$ for each MoE layers $L$, based on the Layer inputs $X$
**for** $l \in L$ **do**
   **for** $e \in E$ **do**
      $S_{[l][e]} \leftarrow |W_{[l][e]}^T| \times ||X_{[l]}||_2$
      $sorted\_idx_{[l][e]} \leftarrow sorted(S_{[l][e]}, \text{descending=True})$
      $pruned\_idx_{[l][e]} \leftarrow sorted\_idx_{[l][1...\text{int}(C_{\text{in}} \times s)]}$
      $expert\_score \leftarrow mean(S_{pruned\_idx_{[l][e]}})$
      $IS_{[l][e]} + = expert\_score$
   **end for**
   $IE_l \leftarrow argmax(IS_{[l][e]})$
**end for**
**return** $IE$

---

Moreover, we propose to quantify the importance of an expert with the proposed score.

**Definition 3.3** (Importance Score). *Let $\mathcal{M}$ denote a sparse mixture-of-expert (SMoE) model with $p$ MoE blocks. In each SMoE block, there are $m$ expert network modules. Each input token selects $k < m$ expert in each SMoE block for computation. Given a dataset $X$ with $n$ tokens, before feeding the input into the MoE blocks, we multiplied each expert's output weights to obtain the Wanda metric values. Next, we sorted the values and applied the sparsity $s$ to eliminate the "unimportant" weights from the Wanda metrics. We then collected the dropped values and take their inverse values as the importance score.*

If the importance score is large, it indicates that the expert has more "important" information in SMoE.

### 3.4 HEAVY-HITTER V.S. IMPORTANCE, A CASE STUDY

In the previous sections, we introduced two perspectives on load balancing in SMoE LLMs: addressing load imbalance by considering both heavy-hitter and less important experts. Using LLaMA-MoE as a case study, we analyzed the PIQA dataset. As shown in Figure 2, the X-axis represents the workload of an expert by the number of allocated tokens, while the Y-axis displays the inverse importance score of an expert (see Definition 3.3). We focused on the performance of the first MoE blocks in LLaMA-MoE. From the figure, it is evident that there is no strong correlation between these two perspectives. An expert with lower importance can still receive a lesser workload. Consequently, we can optimize a pretrained SMoE by reducing the heavy-hitters' workload with additional resources, while simultaneously

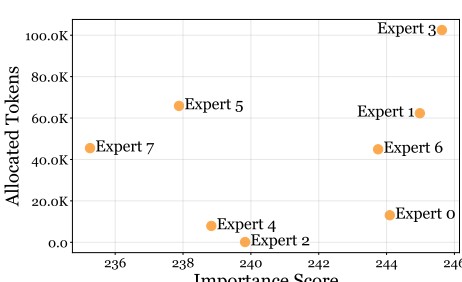

**Figure 2:** Visualization of the allocated tokens vs. inverse importance score for the first MoE blocks at LLaMA-MoE on the PIQA dataset.

decreasing the resources allocated to the least important experts to maintain the overall resource budget.

### 3.5 OUR PROPOSAL: REPLICATE AND QUANTIZE

We propose a novel, plug-and-play strategy. As shown in Figure 1, our approach focuses on identifying and optimizing the utilization of the heaviest and least important experts in the model. Specifically, we introduce a low-cost method to pinpoint the most heavily used experts and then replicate these experts using a lower-bit quantized version to mitigate their load. This replication not only reduces the computational burden but also enhances the parallel processing capabilities of the model, allowing it to handle more complex tasks efficiently. Simultaneously, we quantize the least important experts to ensure the overall model fits within the total memory budget, thereby maintaining efficiency.

Our strategy leverages a dynamic assessment mechanism that continually monitors the performance and usage patterns of each expert during the training and inference phases. By adapting in real-time, our method ensures that the model remains optimized under varying workloads and input complexities. The lower-bit quantization of heavily used experts is performed using advanced techniques that preserve accuracy while significantly reduc-

Table 2: Baseline for the less important experts

|        |            | Ours       | random     | heavy-hitter | raw    |
|--------|------------|------------|------------|--------------|--------|
| LLaMa  | gsm8k      | **0.0417** | 0.0364     | 0.0296       | 0.0425 |
|        | hellaswag  | **0.5179** | 0.5166     | 0.5126       | 0.5414 |
|        | mmlu       | 0.2669     | 0.2639     | **0.2681**   | 0.2781 |
|        | piqa       | **0.7579** | 0.7535     | 0.7524       | 0.7693 |
|        | truthfulqa | 0.2864     | **0.3001** | 0.2864       | 0.2726 |
|        | winogrande | **0.648**  | 0.5572     | 0.6369       | 0.6701 |
| Switch 8 | gsm8k    | 0.0045     | **0.0106** | 0.0038       | 0      |
|        | hellaswag  | **0.2826** | 0.2821     | 0.2825       | 0.2746 |
|        | mmlu       | **0.2295** | 0.2295     | 0.2295       | 0.2295 |
|        | piqa       | **0.5941** | 0.5843     | 0.5925       | 0.5811 |
|        | truthfulqa | 0.3756     | 0.3614     | **0.3792**   | 0.3692 |
|        | winogrande | **0.5454** | 0.5193     | 0.5225       | 0.4964 |

ing memory and computational overhead. This dual approach of replication and quantization creates a balanced distribution of computational resources across the model.

Furthermore, our method includes a feedback loop that reassesses the importance and utilization of experts periodically. This allows the model to adapt to new data patterns and maintain optimal performance over time. By integrating seamlessly with existing models without the need for extensive retraining, our plug-and-play strategy offers a practical solution for enhancing the efficiency and scalability of large neural networks. This approach is particularly beneficial in environments with limited computational resources or in applications requiring real-time processing, where maintaining high performance and efficiency is crucial.

---

**Algorithm 3** Replicate and Quantize

---

**Input:** Model $M$, Experts numbers $E$
**Output:** Replicated and Quantified model $RQ$
**Initialize:** $replicate\_expert \leftarrow Algorithm\ 1(), quantization\_expert \leftarrow Algorithm\ 2()$
$layer\_idx \leftarrow 0$
**for** $layers \in M$ **do**
$\quad re \leftarrow replicate\_expert_{[layer\_idx]}$
$\quad qe \leftarrow quantization\_expert_{[layer\_idx]}$
$\quad layers+ = quant(layers_{[re]})$
$\quad layers_{[qe]} \leftarrow quant(layers_{[qe]})$
$\quad layer\_idx+ = 1$
**end for**
**return** $M$

---

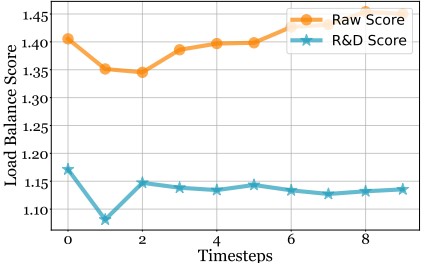

(a) Option1: relay on all of the previous information

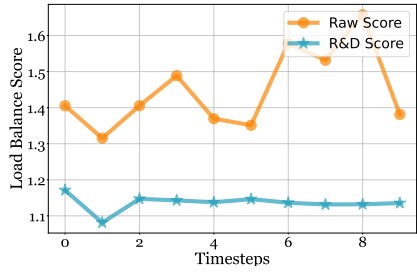

(b) Option2: only relay on the information from the previous one

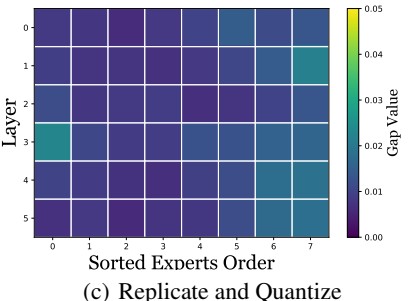

(c) Replicate and Quantize

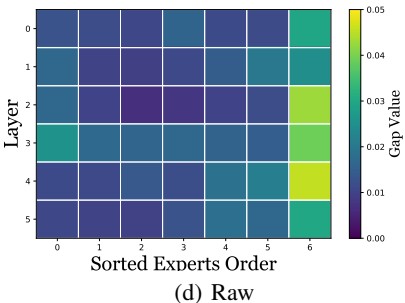

(d) Raw

Figure 3: (Top row) Load balance score comparison between $R\&D$ and Raw for the LLaMa MoE Model in the streaming data. (Bottom row) Comparison between the $R\&D$ and raw gap value among each expert in each layer for the Switch Transformer model.

## 4 EXPERIMENT

In this section, we want to validate the effectiveness of our proposed replicate and quantize strategy. We would like to answer the following research questions:

- **RQ1:** How does the proposed replicate and quantize strategy perform in improving the load imbalance of a pre-trained SMoE?
- **RQ2:** How does the proposed replicate and quantize strategy preserve the predictive performance of SMoE when applied to a pre-trained SMoE?
- **RQ3:** How does the proposed replicate and quantize strategy help SMoE to dynamically adapt to streaming workflow?

### 4.1 DATASET

In this work, we evaluate the performance of our proposed method using a diverse set of benchmark datasets to test different aspects of model performance across a variety of domains and task types. The datasets include Massive Multitask Language Understanding (MMLU), TruthfulQA, Grade School Math (GSM8K), Winogrande, Hellaswag, and Physical Interaction Question Answering (PIQA). For the fine-tuning experiment, we tuned the model using the WikiQA dataset, a public question-answering benchmark focused on the quality of Wikipedia content. During the evaluation of the fine-tuned model, we incorporate the WikiQA dataset into the existing evaluation datasets.

### 4.2 TESTBED

We fine-tuned the Switch Transformer model on one NVIDIA Tesla V100-SXM2-32GB, and all Switch Transformers evaluation experiments were conducted on the NVIDIA 8 Tesla V100-SXM2-32GB U servers. The LLaMa-MoE and DeepSeek-MoE were evaluated on the NVIDIA 4 A100-SXM4-40GB and NVIDIA 8 Quadro RTX 8000. **Evaluation Metric** We use lm-eval-harness to calculate model accuracy for Massive Multitask Language Understanding (MMLU), TruthfulQA, Grade School Math (GSM8K), Winogrande, Hellaswag, and Physical Interaction Question Answering (PIQA).For the Hellaswag datasets, we use a 10-shot approach, while the GSM8K and MMLU datasets use a 5-shot approach. In our evaluation of the fine-tuned Switch Transformer model, we use the generation F1 score as our criterion.

**Workload** Within each layer, we initially calculate the allocation of tokens to each expert, observing the expert who receives the highest number of tokens as the "most heavy expert." Concurrently, we employ the tool "Wanda" to calculate the expert who is considered as the "less significant expert." Next, we apply quantization to the "less important expert" by reducing its value to a half float compared to the original one. Then, we reproduce the "most heavy expert" and apply it to the same quantization process as the "less important expert."

### 4.3 RESULTS

**Answers to RQ1 & RQ2.** We present our results on three pre-trained SMoE LLMs. Given the pre-trained SMoE LLMs with load imbalance, our R&Q method significantly reduces its load imbalance score 3.1 while preserving its predictive performance. Moreover, before that, we have tried to use the different tuning strategies to adjust the router mechanism to solve the load imbalance issues. Clearly, it does not work as we expected, and the part of the strategies emplifies the imbalanced distribution among the different experts. These results answer the first and the second research questions, the proposed R&Q method improves the load imbalance of a pre-trained SMoE while preserving their predictive performance.

**Answers to RQ3** We first feed all the tokens from the MMLU benchmark into the LLaMa MoE model, calculate the tokens that each expert in each layer receives, and then identify the heavy-hitter experts in each layer. Then, we extract those tokens as our streaming data. In this setting, we set the timesteps to 10, so we split the data into 10 portions. For option 1, we apply our method at all time steps when replicating the experts; the current window depends on all of the previous information to determine which expert needs to be replicated; and quantized experts are previously predicted at 10% of the MMLU benchmark. For option 2, the representative information only relies on the former one. In the figure 3, it's obvious that our method can effectively relieve the load imbalance

Table 3: Comparision of the Load Balance and Accuracy

| Model | Dataset | Load Imbalance | | Accuracy | |
|---|---|---|---|---|---|
| | | Raw | R&Q | Raw | R&Q |
| Switch Transformer (8 experts) | GSM8K | 1.9709 | 1.3937 | \ | \ |
| | Truthful QA | 1.4956 | 1.3494 | 0.3692 ± 0.0111 | 0.3640 ± 0.011 |
| | Winogrande | 1.5261 | 1.2146 | 0.4964 ± 0.0141 | 0.4917 ± 0.0141 |
| | Hellaswag | 1.4182 | 1.3623 | 0.2746 ± 0.0045 | 0.2763 ± 0.0045 |
| | MMLU | 1.5405 | 1.2962 | 0.2295 ± 0.0035 | 0.2522 ± 0.0037 |
| | PIQA | 1.5770 | 1.2756 | 0.5811 ± 0.0115 | 0.5832 ± 0.0115 |
| Switch Transformer (16 experts) | GSM8K | 1.4352 | 1.2864 | \ | \ |
| | Truthful QA | 2.0121 | 1.6856 | 0.3738 ± 0.0112 | 0.3694 ± 0.0112 |
| | Winogrande | 2.0063 | 1.5394 | 0.4901 ± 0.014 | 0.4854 ± 0.014 |
| | Hellaswag | 1.9681 | 1.7369 | 0.2857 ± 0.0045 | 0.2872 ± 0.0045 |
| | MMLU | 1.9964 | 1.7067 | 0.2295 ± 0.0035 | 0.2495 ± 0.0036 |
| | PIQA | 2.0355 | 1.6080 | 0.5457 ± 0.0116 | 0.5501 ± 0.0116 |
| LLaMa MoE | GSM8K | 1.3565 | 1.1963 | 0.0349 ± 0.0051 | 0.0364 ± 0.0052 |
| | Truthful QA | 1.2946 | 1.2025 | 0.2726 ± 0.0098 | 0.2730 ± 0.0098 |
| | Winogrande | 1.3925 | 1.2791 | 0.6732 ± 0.0132 | 0.6669 ± 0.0132 |
| | Hellaswag | 1.3047 | 1.2258 | 0.5414 ± 0.0050 | 0.5403 ± 0.0050 |
| | MMLU | 1.3289 | 1.1964 | 0.2797 ± 0.0038 | 0.2797 ± 0.0038 |
| | PIQA | 1.2943 | 1.2366 | 0.7704 ± 0.0098 | 0.7704 ± 0.0098 |
| DeepSeek MoE | GSM8K | 4.8182 | 3.9725 | 0.1562 ± 0.01 | 0.1539 ± 0.0099 |
| | Truthful QA | 2.9672 | 2.1749 | 0.3109 ± 0.0103 | 0.3105 ± 0.0103 |
| | Winogrande | 3.1386 | 2.4475 | 0.7001 ± 0.0129 | 0.7064 ± 0.0128 |
| | Hellaswag | 3.0387 | 2.6403 | 0.5984 ± 0.0049 | 0.5975 ± 0.0049 |
| | MMLU | 3.7359 | 2.8222 | 0.4472 ± 0.0041 | 0.4467 ± 0.0041 |
| | PIQA | 4.0780 | 3.0585 | 0.7884 ± 0.0095 | 0.7905 ± 0.0095 |

issue, even though those data are considered the most severe load imbalance input for this model. Whatever depends on all of the previous information or the only former one, it all shows a clear decrease in the load imbalance score in the line chart. Figure 4 shows the gaps between the ordered expert hitter ratios. The hitter ratio is calculated from the mean value of each expert's allotted tokens, and we sort this ratio to get the difference between the neighboring ones. This heatmap shows that the raw switch transformer model illustrates strong performance by always choosing one expert rather than all of others. But, after applying our method, we found the tokens allocated to each expert to be more balanced; they have a lower gap between the ratios of activation.

## 5 CONCLUSION

In conclusion, large language models with sparse mixture-of-experts (SMoE) architectures have shown empirical success across various tasks. This architecture allows SMoEs to scale up the number of experts without the need for a proportional increase in computational resources, offering an efficient way to improve performance on diverse tasks. However, despite these advantages, SMoE's sparse routing can lead to significant load imbalances among experts, some may become overloaded with too many tasks while others remain underutilized, causing efficiency issues during deployment. Our paper presents a plug-and-play strategy to address this load imbalance. We propose a low-cost method that identifies and replicates the heaviest expert using a lower-bit quantized version while also quantizing the least important expert to maintain the memory budget. We conducted thorough empirical evaluations to validate the effectiveness of this approach. The results indicate that our strategy successfully alleviates the load imbalance issue in SMoE architectures. Furthermore, the R&Q strategy we used resulted in minimal loss of performance, making our method a practical and efficient improvement for the deployment of SMoE models in large-scale systems.

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
