# OpenReview forum: "Replicate and Quantize: A Plug-and-Play Strategy for Load Balancing in Sparse Mixture-of-Experts LLMs"
_ICLR.cc/2025/Conference — Submitted to ICLR 2025_

### Official Review · Reviewer_5vEp · 2024-10-25

**Soundness:** 2
**Presentation:** 3
**Contribution:** 2
**Rating:** 5
**Confidence:** 3

**Summary:**

This work provides a simple yet effective strategy for load balancing in MoE-based LLMs. Specifically, the authors first find the most heavy expert and the less important experts, and (a) replicate and quantize most heavy experts, (b) quantize less important experts. In experiments, the authors have deployed the proposed method on 4 MoE models, achieving comparable results with more balanced load among experts. In conclusion, the proposed model is sound and easy to deploy, while more in-depth evaluations and analyses should be conducted.

**Strengths:**

1)	The proposed idea is simple and sound.
2)	Overall, this work is well-organized and easy to follow.
3)	The authors have tested on various MoE base models.

**Weaknesses:**

1)	The central idea of this work is the replicate and quantize strategy. Firstly, an appropriate ablation study should be conducted to verify the effectiveness of both strategies on the most heavy experts and less important experts. Secondly, whether the selections of most heavy experts and less important experts are essential? If we further quantize other experts while maintaining the activated parameters, what will be the results?
2)	In Section 3.4, the authors merely give the importance score/load results on one task PIQA. Does this phenomenon also exist in other tasks and other MoE blocks? The authors are suggested to give more quantitative indicators (e.g., correlation coefficient on different settings) to support their claims.
3)	In Related work, an in-depth analyses on other load balance methods (and why they do not work well in the experiments) should be given.
4)	In experiments, although the authors claimed that “before that, we have tried to use the different tuning strategies to adjust the router mechanism to solve the load imbalance issues, Clearly, it does not work as we expected, and the part of the strategies emplifies the imbalanced distribution among the different experts”, which strategies are used and the corresponding results should be given. Currently, only using the raw setting as baseline is not sufficient.
5)	The experimental details are insufficient. For instance, the details of adopting the proposed method on DeepSeekMoE should be given. DeepSeekMoE adopts shared and specialized experts, and whether the shared experts are also used for replicate? Moreover, it also multiplies the number of experts, which shares the similar idea of the “replicate” heavy expert part in this work.
6)	The actual inference speed and cost should be given. Do all comparisons share the same activated parameters in inference?
7)	Typos, e.g., Page2, missing reference in the first paragraph.
8)	The scalability of the proposed method is encouraged to be evaluated or discussed.

After rebuttal, I raise the voting to 5.

**Questions:**

Refer to the questions in Weaknesses.

---

> ### Author Response · Authors · 2024-11-22
>
> We thank the reviewers for the comments and suggestions to improve our paper. Please see the following clarification.
>
> ## [W1: ablation study - Sure]
> We provide an ablation study of replication and quantization as follows.
>
>
> | Method                      | Hellaswag           | MMLU               | PIQA               | Truthful QA        | Winogrande         |
> |-----------------------------|---------------------|---------------------|--------------------|--------------------|--------------------|
> | **Switch 8 (only quant)**   | 0.2795             | 0.2295             | 0.5751            | 0.3605            | 0.5138            |
> | **Replicate and quant replicate one** | 0.2749 ± 0.0045 | 0.2498 ± 0.0037 | 0.5811 ± 0.0115 | 0.3635 ± 0.0110 | 0.4917 ± 0.0141 |
> | **Quant ALL**               | 0.2641 ± 0.0044   | 0.2295 ± 0.0035   | 0.5490 ± 0.0116  | 0.3775 ± 0.0112  | 0.5185 ± 0.0140 |
> | **Ours**                    | 0.2763 ± 0.0045   | 0.2522 ± 0.0037   | 0.5832 ± 0.0115  | 0.3640 ± 0.0110  | 0.4917 ± 0.0141 |
> | **Switch 16 (only quant)**  | 0.2820 ± 0.0045   | 0.2295 ± 0.0035   | 0.5577 ± 0.0116  | 0.3914 ± 0.0114  | 0.4964 ± 0.0141 |
> | **Replicate and quant replicate one** | 0.2864 ± 0.0045 | 0.2490 ± 0.0036 | 0.5490 ± 0.0116 | 0.3669 ± 0.0112 | 0.4830 ± 0.0140 |
> | **Quant ALL**               | 0.2595 ± 0.0044   | 0.2295 ± 0.0035   | 0.5582 ± 0.0116  | 0.3721 ± 0.0110  | 0.5091 ± 0.0141 |
> | **Ours**                    | 0.2872 ± 0.0045   | 0.2495 ± 0.0036   | 0.5501 ± 0.0116  | 0.3694 ± 0.0112  | 0.4854 ± 0.0140 |
>
> ## [W2: Does this phenomenon also exist in other tasks and other MoE blocks? - Yeah]
> In Figure 2, our goal is to demonstrate the existence of two distinct dimensions among the experts, which represent our observation framework rather than a phenomenon inherent to the experts themselves. These dimensions provide a structured way to quantify and analyze the functionality of the experts. While PIQA serves as a representative example, we observed that this two-dimensional framework can be applied consistently across other tasks and MoE blocks.
>
> ## [W3: Related Work in load balance - Sure]
> The current load balanced method focuses on the training stage. For the current SMoE model, they have tried to solve their load imbalance issue by introducing the imbalance loss or other methods during training the model. Moreover, there are some current works focus on the gpu utilization rather than the load balance for each experts.
>
> ## [W4: Tuning strategies - we have show their results in Table 1]
>
> ## [W5: The details of adopting the proposed method on DeepSeekMoE should be given - Sure]
> The Deepseek MOE total has 64 experts in each layer: 63 isolated experts and 1 shared expert.  In each routing, the system will select three experts from the 63 isolated experts, always using the shared expert. Therefore, our strategy is exclusively applied to the 63 isolated experts.
>
> ## [W6: Do all comparisons share the same activated parameters in inference? - Yeah, they shared the same activated parameters during the inference]
>
> ## [W7: Format -  Thank you for reminder. We have updated the paper. We will post a revised version with a summary of revision.]
>
> ## [W8: The scalability of the proposed method is encouraged to be evaluated or discussed. - Sure]
> In Figure 3, Panels (a) and (b) illustrate the load balance scores across multiple timesteps under two distinct routing strategies. Panel (a) demonstrates the case where the system utilizes cumulative information aggregated from all prior timesteps, providing a holistic approach to decision-making. Conversely, Panel (b) focuses on a scenario where only information from the immediately preceding timestep is leveraged, showcasing a more localized decision-making approach.
>
> The purpose of this setup is to identify the most important "heavy-hitter" experts from the less important ones. These decisions depend directly on the input data at each timestep. To mimic how data arrives in a real-world streaming scenario, we divided the MMLU dataset into 10 parts and set the timestep count to 10. This approach allows us to simulate the flow of data over time and study how it affects the routing strategy in a manageable and realistic way.

---

> > ### Comment · Reviewer_5vEp · 2024-11-25
> > **After rebuttal**
> >
> > I have read all author rebuttals. The authors have answered lots of my questions on further experiments.
> > I will raise my voting to 5, and suggest that these revisions should be included in the future version.

---

### Official Review · Reviewer_Dy55 · 2024-11-02

**Soundness:** 2
**Presentation:** 2
**Contribution:** 2
**Rating:** 3
**Confidence:** 5

**Summary:**

This paper proposes Replicate and Quantize, an inference-time strategy that aims to mitigate load imbalance on Mixture-of-Expert based Large Language Models. The author claimed that there exist differences between heavy-hitters and important experts in MoE models, and proposed to 1) quantize the least important expert for resource savings, and 2) replicate a quantized version of heavy hitters to reduce load imbalance. Results show that the authors' strategy improved load imbalance without reducing too much performance.

**Strengths:**

- This paper studies an interesting problem, which is the load imbalance of MoE LLMs in inference scenarios.
- The author conducted experiments on various tasks and model types, creating a comprehensive overview of the impact of the proposed strategy on the performance of the model.

**Weaknesses:**

- The paper's claim on its novelty, that the load imbalance of MoE models has not been studied for inference scenarios, is wrong. Plenty of research have focused on the inference scenario, such as [1], [2], [3], [4] and [5], and many of them have provided a thorough characterization of the workload already. Quantization of the MoE model has also been studied in [6]. The author should conduct **a more thorough review on the existing literature** and discuss the difference of this work with these existing ones.

- The Load Imbalance Score defined at Sec 3.1 is an ill-defined metric. The overall load in a dataset is not directly related to the inference performance of the MoE model. It is the load in a certain batch that would have a major impact on the robustness of the model (preventing OOM errors) and latency (of all-to-all communications).

- Algorithm 1 seems to be unnecessary. The search is quite straightforward.

- The authors' proposition, that the heavy-hitters are not necessarily the most important expert, seems to be *rebuked* by the presented data in Fig. 2. See questions below.

- The most important metrics, which are the inference performance of the proposed system (**memory consumption, latency, hardware utilization, and so on**), are not studied in this work. These are the most important reasons one would like to reduce the load imbalance.

- Line 216, "Wanda metric" has been referenced, but only formally defined on line 237.

- The paper is not well formatted. For example:
  - The citations are not correctly adapted to the ICLR format. e.g. Line 107 -- Jacobs et al. Jacobs et al..
  - Missing spaces ",or" on line 115.
  - Missing reference on line 120.
  - Missing spaces ".For" on line 406.

- The authors have not provided any code or reproducibility statement.


[1] Huang, Haiyang, et al. "Towards MoE Deployment: Mitigating Inefficiencies in Mixture-of-Expert (MoE) Inference." arXiv preprint arXiv:2303.06182 (2023).

[2] Gale, Trevor, et al. "Megablocks: Efficient sparse training with mixture-of-experts." Proceedings of Machine Learning and Systems 5 (2023): 288-304.

[3] Kong, Rui, et al. "Serving MoE Models on Resource-constrained Edge Devices via Dynamic Expert Swapping." arXiv preprint arXiv:2308.15030 (2023).

[4] Li, Jiamin, et al. "Accelerating distributed MoE training and inference with lina." 2023 USENIX Annual Technical Conference (USENIX ATC 23). 2023.

[5] Hwang, Ranggi, et al. "Pre-gated moe: An algorithm-system co-design for fast and scalable mixture-of-expert inference." 2024 ACM/IEEE 51st Annual International Symposium on Computer Architecture (ISCA). IEEE, 2024.

[6] Kim, Young Jin, Raffy Fahim, and Hany Hassan Awadalla. "Mixture of Quantized Experts (MoQE): Complementary Effect of Low-bit Quantization and Robustness." arXiv preprint arXiv:2310.02410 (2023).

**Questions:**

- Fig. 2 shows that the most important expert is the expert that receives the most tokens. It seems like it is rejecting, instead of confirming the authors' proposition, that the heavy-hitters are not necessarily the most important expert. Wouldn't quantizing expert 3, the heavy hitter in this case lead to performance degradation?

- How does the router adapt to the case where the most important expert is replicated? Will it evenly distribute its tokens to each GPU device?

- How are the expert loaded on the GPU? Are the other experts completely unaffected?

---

> ### Author Response · Authors · 2024-11-22
>
> We thank the reviewers for the comments and suggestions to improve our paper. Please see the following clarification.
>
> ## [Q1: The paper's claim on its novelty, that the load imbalance of MoE models has not been studied for inference scenarios, is wrong. - We will discuss those papers as follows]
>
> ---
> ### [Towards MoE Deployment: Mitigating Inefficiencies in Mixture-of-Expert (MoE) Inference.](https://openreview.net/pdf?id=stXtBqyTWX)
> **Focus**: load balance for each GPU, not the experts, and they do not release their after-accept by Nips 2024, and the description in the paper is too vague for us to replicate their methods.
>
> ### [MEGABLOCKS: EFFICIENT SPARSE TRAINING WITH MIXTURE-OF-EXPERTS](https://arxiv.org/pdf/2211.15841)
>
> **Focus**: Sparse training, not inference.
>
> ---
>
> ### [SwapMoE: Serving Off-the-shelf MoE-based Large Language Models with Tunable Memory Budget](https://arxiv.org/pdf/2308.15030)
>
> **Focus**: Inference on edge devices with memory constraints. Dynamically load, activate, and swap experts during inference. From the system aspects
>
> ---
>
> ### [Accelerating Distributed MoE Training and Inference with Lina](https://www.usenix.org/conference/atc23/presentation/li-jiamin)
>
> **Focus**: Both training and inference.
> **Load Balancing**: If an expert is overloaded, another expert from the top-k set (on the same device) is selected.
>
> ---
>
> ### [Pre-gated MoE: An Algorithm-System Co-Design for Fast and Scalable Mixture-of-Expert Inference](https://arxiv.org/pdf/2308.12066)
>
> **Focus**: Add a pre-gating mechanism, which modifies the model architecture.
>
> ---
>
> ### [Mixture of Quantized Experts (MoQE): Complementary Effect of Low-bit Quantization and Robustness](https://arxiv.org/pdf/2310.02410)
>
> **Focus**: Memory optimization via quantization. They mainly conduct thorough experiments to show the experts weight is more robust than we expect.
>
> ---
> ## [W2: The Load Imbalance Score defined at Sec 3.1 is an ill-defined metric. - It's good ]
> We measure the load distribution across experts for the entire dataset because a batch is typically a uniform sample from the dataset. Therefore, using the dataset-level measurement provides a reasonable approximation.
>
> ## [W4 & Q1: Fig. 2 shows that the most important expert is the expert that receives the most tokens. - The symptom showing the experts inner function does not influence our strategy.]
> In Figure 2, our goal is to demonstrate the existence of two distinct dimensions among the experts, highlighting that their functionality can be analyzed from these two different perspectives. While it is true that in some cases, the most important expert may also be the heaviest one, this does not hinder our method to quantize the less important experts and replicate the heavier ones. While certain experts may excel in multiple aspects, this does not negate the presence of distinct functional dimensions that allow us to manage and optimize the experts accordingly.
>
> ## [W5 & W6: Thanks for your suggestion, we will modify it!]
>
> ## [Q2: How does the router adapt to the case where the most important expert is replicated? Will it evenly distribute its tokens to each GPU device?  - Randomly and  Not necessary]
> When the router adapts to the case where the most important expert is replicated, it randomly assigns tokens to one of the replicated experts for processing.
>
> Based on our experiments, splitting the workload of a heavy-hitter expert between just two replicas significantly alleviates the load imbalance. It is not necessary to distribute the workload across all GPUs to achieve this improvement.
>
> ## [Q3: How are the expert loaded on the GPU? Are the other experts completely unaffected? - Not affected]
> In the quantized version, the compressed weights are loaded into the GPU as low-bit representations. This allows the quantized experts to occupy significantly less memory, enabling their efficient deployment on the GPU.
>
> Other experts are unaffected by this process as their weights remain unchanged and continue to operate in their original precision.

---

> > ### Comment · Reviewer_Dy55 · 2024-11-23
> >
> > I thank the authors for their detailed rebuttal. While some of my concerns have been addressed, the most critical issues remain unresolved, which are W5 and W8, respectively.
> > - W5: The most important metrics, which are the inference performance of the proposed system (memory consumption, latency, hardware utilization, and so on), are not studied in this work. These metrics are crucial for justifying the significance of reducing load imbalance. Without measurements of these quantities, it is difficult to assess the practical improvements brought by the proposed strategy and its impact on the field. I strongly encourage the authors to outline a concrete plan for evaluating these metrics and, if feasible, provide preliminary results.
> >
> > - W8: The authors have not provided any code or reproducibility statement. The official implementations of both Llama-MoE and DeepSeek-MoE, when deployed with Huggingface Transformers, only support native tensor parallelism. The lm-eval-harness framework also do not provide an expert parallelism implementation. Which open-sourced MoE framework are you building upon? Did you implement the framework from scratch?
> >
> > Other questions include:
> > - W1. I appreciate the authors’ effort in expanding the review of related works. And I think one work that is closely related is [7]. The Dynamic Shadowing Strategy in Section 4.1 of [7] appears quite similar to the proposed replicate strategy, aside from the quantization component. The strategy does not have any training-only component, thus can be applied to inference scenarios. I encourage the authors to clearly differentiate their approach from [7] to establish the novelty of their contribution. The author should also discuss [6] in depth to answer the question in W4 & Q1 (see below).
> >
> > - W2. Hardware-level metrics are directly impacted by batch-level statistics, making dataset-level metrics less relevant in this context. I would expect the load imbalance score to increase when evaluated at the batch level, which would enhance your argument.
> >
> > - W4 & Q1. The data provided in the paper does not convincingly demonstrate the significance of the difference between the two metrics. As stated on line 096, only the most important expert (which happens to be the heavy-hitter) and the least important experts are quantized in the experiments. The overlap between heavy-hitter and most important expert limits the proposed metric's ability to make a substantial difference in preserving accuracy. A more compelling example illustrating where these experts diverge would strengthen the argument. Additionally, the paper should address the broader question: what is the accuracy loss when the model is fully quantized? From [6], it seems like quantizing all the experts only incurs a negligible accuracy loss.
> >
> > [7] He, Jiaao, et al. "Fastermoe: modeling and optimizing training of large-scale dynamic pre-trained models." Proceedings of the 27th ACM SIGPLAN Symposium on Principles and Practice of Parallel Programming. 2022.

---

> > > ### Author Response · Authors · 2024-11-27
> > >
> > > We thank the reviewers for the comments and suggestions to improve our paper. Please see the following clarification.
> > >
> > > | Model & Configuration            | Hellaswag            | MMLU                | PIQA                | Truthful QA         | Winogrande         |
> > > |----------------------------------|----------------------|---------------------|---------------------|---------------------|---------------------|
> > > | **Switch 8**                     |                      |                     |                     |                     |                     |
> > > | Quant All Experts                 | 0.2641 ± 0.0044     | 0.2295 ± 0.0035     | 0.5490 ± 0.0116     | 0.3775 ± 0.0112     | 0.5185 ± 0.014      |
> > > | R&D                               | 0.2763 ± 0.0045     | 0.2522 ± 0.0037     | 0.5832 ± 0.0115     | 0.3640 ± 0.0110     | 0.4917 ± 0.0141     |
> > > | **Switch 16**                     |                      |                     |                     |                     |                     |
> > > | Quant All Experts                 | 0.2595 ± 0.0044     | 0.2295 ± 0.0035     | 0.5582 ± 0.0116     | 0.3721 ± 0.0110     | 0.5091 ± 0.0141     |
> > > | R&D                               | 0.2872 ± 0.0045     | 0.2495 ± 0.0036     | 0.5501 ± 0.0116     | 0.3694 ± 0.0112     | 0.4854 ± 0.014      |
> > > ## [W5: Inference Performance: Sure]
> > > Our method ensures memory usage does not increase even when duplicating new experts. After duplication, we quantize the new experts and further quantize less-important ones to half precision.
> > >
> > > Regarding inference speed, in each MoE layer, the inference time is determined by the slowest expert, which handles the largest number of tokens. Below, we provide an example to illustrate this behavior.
> > >
> > > **Premise:**
> > >
> > > - **Number of Experts:** 4
> > > - **Number of Tokens:** 80
> > > - **Processing Capacity per Expert:** Each expert processes \( \frac{80}{4} = 20 \) samples per unit time (T).
> > >
> > > ---
> > >
> > > **Original Routing Results:**
> > >
> > > - **Experts' Token Allocation:** [18, 30, 25, 7]
> > > - **Inference Time (Slowest Expert):**
> > >   Expert 1 handles 30 tokens:
> > >   \[
> > >   $\frac{30}{20} = 1.5 \, T$
> > >   \]
> > >
> > > ---
> > >
> > > **After Rerouting & Quantization (R & Q):**
> > >
> > > - **Experts' Token Allocation:** [18, [17, 13], 25, 7]
> > > - **Inference Time (Slowest Expert):**
> > >   Expert 2 handles 25 tokens:
> > >   \[
> > >   $\frac{25}{20} = 1.25 \, T$
> > >   \]
> > >
> > > By applying our method, the inference time is reduced, resulting in a speed-up of 0.25.
> > >
> > >
> > > ## [W8: Code Release - Sure]
> > > Thank you for your question. Our current implementation is a simulation code. Depending on the final decision for the paper, we plan to release the simulation code along with detailed documentation to support reproducibility.
> > >
> > > ## [W1: Paper [7]]
> > > They focus on the training strategy and duplicate the expert to other free GPU to release this issue during training.
> > >
> > > ## [W2: batch-level eval - yeah, we have conducted experiments on it]
> > > | Task           | Batch Size 1 | Batch Size 32 |
> > > |----------------|--------------|---------------|
> > > | GSM8K          | 1.9709       | 3.4523        |
> > > | Truthful QA    | 1.4956       | 3.2967        |
> > > | Winogrande     | 1.5261       | 2.7249        |
> > > | Hellaswag      | 1.4182       | 2.5058        |
> > > | MMLU           | 1.5405       | 3.9897        |
> > > | PIQA           | 1.5770       | 3.8401        |
> > >
> > > ## [W4 & Q1: Two Metrics]
> > > In this method, we observe that some experts handle significantly more tokens than others, leading to load imbalance during the inference stage. Instead of randomly redistributing tokens to other experts, which might compromise accuracy, we propose a strategy called "replicating the heavy-hitter expert." This involves creating a duplicate of the heavy-hitter expert, with tokens randomly assigned to one of the replicas. Ideally, this approach reduces the workload by half. However, it increases memory usage. To address this, we quantize both the replicated expert and the less important experts to half-precision, maintaining the overall memory footprint. Our findings show that half-precision quantization effectively preserves accuracy. Randomly quantizing any expert could negatively impact real-world performance or specific tasks, so we prioritize quantizing the less important experts instead.

---

> > > > ### Author Response · Authors · 2024-11-29
> > > >
> > > > Hello Reviewer Dy55,
> > > >
> > > > We hope you had a wonderful Thanksgiving! We are checking in on our previous comment to see if you've had a chance to read it, and we'd be grateful if you could have a moment to review it.
> > > >
> > > > We address your concerns in our rebuttal.
> > > >
> > > > * The inference time will decrease, and the memory usage will remain unchanged.
> > > > * We will release our simulation code depending on the final decision.
> > > > * Paper [7] focuses on the training strategy.
> > > > * Clarify our metric—the overlap between the most important experts and heavy-hitter experts; do not limit our metric ability to preserve accuracy as shown in our Table 3,  and we conducted the experiments you mentioned here.
> > > > * We demonstrated a more pronounced load imbalance in a larger batch, and our strategy can still effectively address this phenomenon.
> > > >
> > > > | Model & Configuration            | Hellaswag            | MMLU                | PIQA                | Truthful QA         | Winogrande         |
> > > > |----------------------------------|----------------------|---------------------|---------------------|---------------------|---------------------|
> > > > | **Switch 8**                     |                      |                     |                     |                     |                     |
> > > > | Quant All Experts                 | 0.2641 ± 0.0044     | 0.2295 ± 0.0035     | 0.5490 ± 0.0116     | 0.3775 ± 0.0112     | 0.5185 ± 0.014      |
> > > > | R&D                               | 0.2763 ± 0.0045     | 0.2522 ± 0.0037     | 0.5832 ± 0.0115     | 0.3640 ± 0.0110     | 0.4917 ± 0.0141     |
> > > > | Quant All Model                   | 0.2927 ± 0.0045     | 0.2295 ± 0.0035     | 0.5952 ± 0.0115     | 0.3706 ± 0.0111     | 0.5091 ± 0.0141     |
> > > > | **Switch 16**                     |                      |                     |                     |                     |                     |
> > > > | Quant All Experts                 | 0.2595 ± 0.0044     | 0.2295 ± 0.0035     | 0.5582 ± 0.0116     | 0.3721 ± 0.0110     | 0.5091 ± 0.0141     |
> > > > | R&D                               | 0.2872 ± 0.0045     | 0.2495 ± 0.0036     | 0.5501 ± 0.0116     | 0.3694 ± 0.0112     | 0.4854 ± 0.014      |
> > > > | Quant All Model                   | 0.2768 ± 0.0045     | 0.2295 ± 0.0035     | 0.5539 ± 0.0116     | 0.3726 ± 0.0112     | 0.4901 ± 0.014      |
> > > >
> > > > Batch Size 32
> > > > | Task           | R & Q | Raw |
> > > > |----------------|--------------|---------------|
> > > > | GSM8K          | 3.1689       | 3.4523        |
> > > > | Truthful QA    | 2.2348       | 3.2967        |
> > > > | Winogrande     | 1.5515       | 2.7249        |
> > > > | Hellaswag      | 1.7974       | 2.5058        |
> > > > | MMLU           | 2.7664       | 3.9897        |
> > > > | PIQA           | 3.6789       | 3.8401        |
> > > >
> > > > If you have any updates or thoughts, we’d greatly appreciate your feedback. Please let us know if there’s anything we can clarify or assist with.
> > > >
> > > > Best regards,
> > > >
> > > > Authors

---

### Official Review · Reviewer_CGsx · 2024-11-02

**Soundness:** 2
**Presentation:** 3
**Contribution:** 2
**Rating:** 5
**Confidence:** 4

**Summary:**

This paper introduces a plug-and-play approach (R&Q) for addressing load imbalance in Sparse Mixture-of-Experts models to improve computational efficiency without retraining. R&Q literally replicates heavily used experts in a quantized form and quantizes less important experts to maintain memory efficiency. Minimal impact on performance.

**Strengths:**

S1. No retraining is required.

S2. Near-original accuracy (at least on classification / MCQ tasks)

S3. I like how they distinguished between heavy-hitter experts and important experts, which could easily be confused as the same. They also conducted experiments to show that these concepts are distinct, although there is some correlation between them.

**Weaknesses:**

W1. Weak presentation of algorithms and figures. Some figures lack any caption or explanation. In Algorithm 1, the choice of variable names is awkward and confusing. For example, l(x), count(expert_chosen), argmax(expert_num), EC, etc., need to be clarified with better names.

W2. Weak baseline. The baseline experiments were only conducted within their framework (ours vs. random vs. heavy-hitter). They lack comparisons with other techniques that address load balancing.

W3. Weak empirical analysis on computational efficiency gain. While their experiments show that R&Q improves load balancing compared to naive techniques, they don't demonstrate how this improvement directly translates to reduced inference latency. This is critical because the use of quantization could often slow down inference.

W4. Weak empirical analysis on more challenging tasks, such as generation tasks (e.g., perplexity, code generation, MT-Bench, etc.).

**Questions:**

Q1. Error on Page 6, line 288: Are the X-axis and Y-axis labels inverted?

Q2. Should R&Q identify heavy-hitter and important experts for each individual task, or can the identified experts be reused across tasks? The motivation behind this question is that heavy-hitters may vary depending on task characteristics. For example, experts 1-3 might be heavy-hitters for task A, while different experts could be heavy-hitters for task B.

Q3. While resolving load imbalance could theoretically improve computational efficiency, how does R&Q empirically achieve this efficiency gain? Could it actually slow down inference latency due to the quantized experts? I’m asking this because the experiment section lacks an empirical analysis of memory and latency improvements. A strong answer to this question would require empirical results.

Q4. Would R&Q maintain performance on more challenging tasks, such as generation tasks (e.g., perplexity, code generation, MT-Bench, etc.)?

---

> ### Author Response · Authors · 2024-11-22
>
> We thank the reviewers for the comments and suggestions to improve our paper. Please see the following clarification.
>
> ## [W1: Weak presentation of algorithms and figures. Some figures lack any caption or explanation. - Thanks! We will complement them! ]
>
> ### Algorithm 1: Identify Heavy-Hitter Experts
> **Purpose**: This algorithm identifies the most frequently selected "heavy-hitter" experts for each Mixture-of-Experts (MoE) layer based on input tokens and their routing.
>
> #### Inputs:
> - **Tokens**: `input_tokens` - A list of input tokens.
> - **Experts**: `num_experts` - The number of experts in each MoE layer.
> - **Layers**: `num_layers` - The number of MoE layers.
> - **Token Count**: `num_tokens` - The number of tokens to be processed.
>
> #### Outputs:
> - **Heavy Experts**: `heavy_hitters` - A list where each entry corresponds to the most selected expert in each layer.
>
> ---
>
> #### Algorithm:
>
>
> 1. Initialize: heavy_hitters ← list of size `num_layers`
>
> 2. For each layer `layer_index` in range `num_layers` do:
>
>    a. Initialize: expert_selection ← empty list
>
>    b. For each token `token` in `input_tokens` do:
>
>       i. selected_expert ← route_token_to_expert(token, layer_index)
>
>       ii. Append selected_expert to expert_selection
>
>    c. Compute expert_frequencies ← count_frequency(expert_selection)
>
>    d. heavy_hitter ← find_expert_with_max_frequency(expert_frequencies)
>
>    e. Store heavy_hitter in heavy_hitters[layer_index]
>
> 3. Return heavy_hitters
>
> ## [Q1: Error on Page 6, line 288: Are the X-axis and Y-axis labels inverted? - Thanks]
> Change to “As shown in Figure 2, the Y-axis represents the workload of an expert by the number of allocated tokens, while the X-axis displays the inverse importance score of an expert (see Definition 3.3). “
>
> ## [Q2: Should R&Q identify heavy-hitter and important experts for each individual task, or can the identified experts be reused across tasks? - Of course]
> They should be identified for each task; as you mentioned, each expert may have a different performance. Moreover, we have evidence that for each task, we can only utilize 0.1% of the data to identify the heavy-hitter experts and less important experts. We then apply the R&Q strategy to the remaining data, focusing on those experts who continue to perform well.
>
> ## [W2: Weak baseline. - The baseline you mentioned is the one for choosing the less-important experts, not the one for load balance; this one should refer to Table 1.]
> The current load balanced method focuses on the training stage. For the current SMoE model, they have tried to solve their load imbalance issue by introducing the imbalance loss or other methods during training the model. So in our Table 3, the raw load balance scores are the model already equipped with their load balanced method.
>
> ## [W3 & Q3: Weak empirical analysis on computational efficiency gain. - The quantization did slow down inference]
> Theoretically, our strategy will not increase the memory because when we duplicate the heavy-hitter experts, we also quantize the duplicate one to a half precision, then we quantize the less important one to a half precision.
>
> ## [W4 & Q4: Updated the more challenging tasks - Sure!]
> ## Results on other generation task:
> ## TruthfulQA Results
>
> | Model          | Type | bleu_acc | bleu_diff | rougeL_acc |
> |----------------|------|----------|-----------|------------|
> | **DeepSeek MoE** | RQ   | 0.3133   | -7.7523   | 0.2876     |
> |                 | Raw  | 0.3121   | -7.7476   | 0.2901     |
> | **Llama MoE**   | RQ   | 0.2925   | -8.1597   | 0.2546     |
> |                 | Raw  | 0.2913   | -8.5101   | 0.2436     |
>
> ## Load Balance Results
>
> | Dataset                | Type | Value                       | Type | Metric            | Value   |
> |------------------------|------|-----------------------------|------|-------------------|---------|
> | **codexglue_code2text**| Raw  | 2.6119107948506235          | Raw  | smoothed_bleu_4   | 1.5517  |
> |                        | RQ   | 1.9426032317889572          |  RQ  |                   | 1.5463  |
> | **coqa**               | Raw  | 2.0412320534522754          | Raw   | f1                | 0.0104  |
> |                        | RQ   | 1.6487805751851754          |RQ    |                   | 0.0106  |
> | **wikitext**           | Raw  | 2.090017361111111           | Raw   | byte_perplexity   | 17.2096 |
> |                        | RQ   | 1.6235243055555557          |  RQ    |                   | 17.2826 |
> |                        |      |                             | Raw   | bits_per_byte     | 4.1051  |
> |                        |      |                             |RQ|                   | 4.1112  |

---

> > ### Comment · Reviewer_CGsx · 2024-11-26
> >
> > Thank you to the authors who provided additional context. It sounds good that the load balancing helps with more complicated tasks. However, it's still not empirically clear how their method actually translates to speedups or memory efficiency, which is crucial to this technique's usability.

---

> > > ### Author Response · Authors · 2024-11-27
> > >
> > > Our method ensures memory usage does not increase even when duplicating new experts. After duplication, we quantize the new experts and further quantize less-important ones to half precision.
> > >
> > > Regarding inference speed, in each MoE layer, the inference time is determined by the slowest expert, which handles the largest number of tokens. Below, we provide an example to illustrate this behavior.
> > >
> > > ---
> > >
> > > ### Example for Each MoE Layer
> > >
> > > **Premise:**
> > >
> > > - **Number of Experts:** 4
> > > - **Number of Tokens:** 80
> > > - **Processing Capacity per Expert:** Each expert processes $\( \frac{80}{4} = 20 \)$ samples per unit time (T).
> > >
> > > ---
> > >
> > > **Original Routing Results:**
> > >
> > > - **Experts' Token Allocation:** [18, 30, 25, 7]
> > > - **Inference Time (Slowest Expert):**
> > >   Expert 1 handles 30 tokens:
> > >   \[
> > > $  \frac{30}{20} = 1.5 \, T$
> > >   \]
> > >
> > > ---
> > >
> > > **After R & Q:**
> > >
> > > - **Experts' Token Allocation:** [18, [17, 13], 25, 7]
> > > - **Inference Time (Slowest Expert):**
> > >   Expert 2 handles 25 tokens:
> > >   \[
> > > $  \frac{25}{20} = 1.25 \, T$
> > >   \]
> > >
> > > ---
> > >
> > > ### Result:
> > >
> > > By applying our method, the inference time is reduced, resulting in a speed-up of \( 0.25 \, T \).
> > >
> > > ---

---

> > > > ### Author Response · Authors · 2024-11-29
> > > >
> > > > Hello Reviewer CGsx,
> > > >
> > > > We hope you had a wonderful Thanksgiving! We are checking in on our previous comment to see if you've had a chance to read it, and we'd be grateful if you could have a moment to review it.
> > > >
> > > > We address your concerns in our rebuttal.
> > > > * **Applying our strategy will not increase memory usage.**
> > > > * **The inference time will decrease, as we show in our load balance results table, and we also provide an example to show how it works.**
> > > >
> > > >
> > > > If you have any updates or thoughts, we’d greatly appreciate your feedback. Please let us know if there’s anything we can clarify or assist with.
> > > >
> > > > Best regards,
> > > >
> > > > Authors

---

> > > > > ### Comment · Reviewer_Dy55 · 2024-11-30
> > > > >
> > > > > Happy Thanksgiving, and thank you for your efforts in addressing my earlier comments. Unfortunately, I find that the response has again left my key concerns unaddressed and, in some cases, has reinforced my doubts about the paper. Below, I outline the specific issues that remain unresolved:
> > > > >
> > > > > - W5: The primary benefit of the proposed strategy hinges on its ability to deliver tangible improvements in real-world systems. However, the results presented, such as the load imbalance score, are preliminary and speculative. For a high-impact venue like ICLR, we expect the authors to develop at least a minimum viable product and provide substantial experimental evidence supporting their claims. The proposed strategy introduces potential complications that could hinder practical application:
> > > > >
> > > > >   -  Randomly assigning tokens destined for the same expert to different GPUs could cause memory fragmentation and increase memory allocation overhead. Additionally, this approach may necessitate more complex all-to-all communication patterns, which must be taken care of in implementation.
> > > > >
> > > > >   - Uneven distribution of experts across GPUs may lead to bottlenecks, as experts with heavier workloads may introduce overheads that negate potential gains.
> > > > >
> > > > >   - The type casting and mixed-precision processing proposed could further introduce computational overhead.
> > > > >
> > > > > To convincingly establish the benefits of this work, it is crucial for the authors to implement the full mechanism and address these practical challenges experimentally.
> > > > >
> > > > > - The illustrative example presented in the paper contains notable flaws:
> > > > >   1. In the example, tokens for Expert 1 are split into two batches: 17 and 13. It is unclear where the batch of 13 tokens is directed. If it is sent to GPU0, the total workload becomes 31 tokens, exceeding the original maximum of 30 tokens. Since expert quantization requires pre-allocation, such decisions cannot be deferred until the gating function is applied.
> > > > >
> > > > >   2. Previous research, as well as my own observations, indicate that the time required to process tokens does not scale linearly with the token count. This is especially evident when the number of tokens per expert is small, as is the case here.
> > > > >
> > > > > - W8: The lack of a real implementation exacerbates concerns regarding the work's practical advantages. Notably, official implementations of Llama-MoE and DeepSeekMoE on Huggingface promote tensor parallelism, effectively balancing loads and eliminating expert imbalance issues. Since these models do not rely on expert parallelism, the problem addressed by the proposed strategy does not arise in these cases. Without further justification, the relevance of this work remains questionable.
> > > > >
> > > > > - W1: As I mentioned in my previous comment, the proposed "replicate strategy" bears strong similarity to the "Dynamic Shadowing Strategy" described in Section 4.1 of [7], differing primarily in the inclusion of quantization, which is also a common strategy (see below for my concerns on quantization). Given that the strategy lacks training-specific components, it can already be applied to inference scenarios. Changing the application domain from training to inference does not, in my view, constitute sufficient novelty for an ICLR paper.
> > > > >
> > > > > - W4 & Q1: The authors state that "Randomly quantizing any expert could negatively impact real-world performance or specific tasks, so we prioritize quantizing the less important experts instead." However, this claim is not substantiated with evidence in the paper. Furthermore, [6] reports contradictory findings on a related task, making it difficult to accept this assertion without additional justification.

---

### Official Review · Reviewer_UpDR · 2024-11-03

**Soundness:** 2
**Presentation:** 2
**Contribution:** 2
**Rating:** 6
**Confidence:** 2

**Summary:**

This paper introduces a novel strategy called "Replicate and Quantize" for addressing load balancing issues in Sparse Mixture-of-Experts (SMoE) models. The authors systematically analyze the performance and functionality of each expert and introduce a metric to evaluate load balance. They propose a dynamic plug-and-play strategy that is both trainingless and near-lossless, effectively resolving load balancing problems by replicating heavily used experts with lower-bit quantized versions and quantizing the least important experts to fit within the memory budget. Empirical results demonstrate that this approach significantly reduces load imbalance with minimal impact on model performance.

**Strengths:**

1) The "Replicate and Quantize" strategy is a novel approach that dynamically addresses load imbalance in SMoE models without requiring extensive retraining.
﻿
2)  The proposed strategy is plug-and-play, making it easy to integrate with existing models and practical for real-world applications.

**Weaknesses:**

1) The paper lacks detailed implementation specifics, such as the exact quantization methods and hyperparameters used.

2) There is a need for more extensive ablation studies to isolate and demonstrate the contributions of the replication and quantization components individually.

**Questions:**

1) Can you provide more detailed implementation details, including the specific quantization techniques and hyperparameters used in your experiments, to facilitate reproducibility?

2) Could you conduct additional ablation studies to demonstrate the individual contributions of the replication and quantization components in your proposed method?

3) How does your method perform under different levels of model sparsity and varying numbers of experts in the SMoE models?

---

> ### Author Response · Authors · 2024-11-22
>
> We thank the reviewers for the comments and suggestions to improve our paper. Please see the following clarification.
>
> ## [W1&Q1: Implementation Details - Sure!]
> ## Hyperparameters
>
> | Parameter             | Value      |
> |-----------------------|------------|
> | **Learning Rate**     | 5e-5       |
> | **Train Epochs**      | 10         |
> | **Training Batch Size** | 8         |
> | **Eval Batch Size**    | 16         |
> | **Weight Decay**      | 0.01       |
> | **Optimizer**         | AdamW      |
>
> Quantization techniques:
> For Switch Transformer they load in float32 so we use the half quant with the float16
> For Llama and Deepseek which load in float16, so we used 8 bit quantization refer (https://arxiv.org/abs/2208.07339)
>
> ## [W2&Q2: Additional ablation studies to demonstrate the individual contributions of the replication and quantization - Sure!]
> We provide an ablation study of replication and quantization as follows.
>
>
> | Model                              | Hellaswag            | MMLU               | PIQA               | Truthful QA        | Winogrande        |
> |------------------------------------|----------------------|--------------------|--------------------|--------------------|-------------------|
> | **Switch 8**                       |                      |                    |                    |                    |                   |
> | Only Quant Less-Important Experts  | 0.2795 ± 0.0045      | 0.2295 ± 0.0035    | 0.5751 ± 0.0115    | 0.3605 ± 0.0110    | 0.5138 ± 0.014    |
> | Replicate and Quant replicate one  | 0.2749 ± 0.0045      | 0.2498 ± 0.0037    | 0.5811 ± 0.0115    | 0.3635 ± 0.0110    | 0.4917 ± 0.0141   |
> | Quant All Experts                  | 0.2641 ± 0.0044      | 0.2295 ± 0.0035    | 0.5490 ± 0.0116    | 0.3775 ± 0.0112    | 0.5185 ± 0.014    |
> | R&D                                | 0.2763 ± 0.0045      | 0.2522 ± 0.0037    | 0.5832 ± 0.0115    | 0.3640 ± 0.011     | 0.4917 ± 0.0141   |
> | **Switch 16**                      |                      |                    |                    |                    |                   |
> | Only Quant Less-Important Experts  | 0.2820 ± 0.0045      | 0.2295 ± 0.0035    | 0.5577 ± 0.0116    | 0.3914 ± 0.0114    | 0.4964 ± 0.0141   |
> | Replicate and Quant replicate one  | 0.2864 ± 0.0045      | 0.2490 ± 0.0036    | 0.5490 ± 0.0116    | 0.3669 ± 0.0112    | 0.483 ± 0.014     |
> | Quant All Experts                  | 0.2595 ± 0.0044      | 0.2295 ± 0.0035    | 0.5582 ± 0.0116    | 0.3721 ± 0.0110    | 0.5091 ± 0.0141   |
> | R&D                                | 0.2872 ± 0.0045      | 0.2495 ± 0.0036    | 0.5501 ± 0.0116    | 0.3694 ± 0.0112    | 0.4854 ± 0.014    |
>
>
>
> The replication is for load balancing and it maintains the model performance.
> The quantization of less important expert ensures us to fit the total memory budget while maintaing the overall model performance.
>
> ## [Q3: Studies on different levels of model sparsity- We include SMoEs with various sparsity.]
>
> We have conducted experiment in the paper with the following models with different sparsity.
>
> **Switch Transformer 8 experts (Sparsity 1/8):**
> total have 8 experts in each layer, only choose 1 expert for each token
>
> **Switch Transformer 16 experts  (Sparsity 1/16):**
> total have 16 experts in each layer, only choose 1 expert for each token.
>
> **Llama MoE 8 experts (Sparsity 2/8):**
> total have 8 experts in each layer, choose 2 experts for each token
>
> **Deepseek MoE (Sparsity 1+3/63):**
> total has 64 experts in each layer: 63 isolated experts and 1 shared expert.  In each routing, the system will select three experts from the 63 isolated experts, always using the shared expert. Therefore, our strategy is exclusively applied to the 63 isolated experts.

---

> > ### Author Response · Authors · 2024-11-29
> >
> > Hello Reviewer UpDR,
> >
> > We hope you had a wonderful Thanksgiving! We are checking in on our previous comment to see if you've had a chance to read it, and we'd be grateful if you could have a moment to review it.
> >
> > We address your concerns in our rebuttal.
> > * **Provide clear implementation details.**
> > * **Add ablation studies to demonstrate the individual contributions of the replication and quantization.**
> > * **Clarify that the model test in our experiments has different levels of sparsity.**
> >
> > If you have any updates or thoughts, we’d greatly appreciate your feedback. Please let us know if there’s anything we can clarify or assist with.
> >
> > Best regards,
> >
> > Authors

---

> ### Comment · Reviewer_UpDR · 2024-12-03
> **Thank you**
>
> Thanks for the detailed response! I have adjusted my score given the reviews and the replies to reflect these improvements.

---

### Author Response · Authors · 2024-12-04
**A summary of rebuttal**

We sincerely thank the reviewers and area chairs for their thoughtful feedback and engaging discussions on this paper. To support clearer summarization and discussion, we have included a concise, one-step overview of our work.



## **Background, Motivation, and Contributions**

### **Task Motivation**

The inference performance of Sparse Mixture-of-Experts (SMoE) models is hindered by load imbalance, where certain "heavy-hitter" experts process disproportionately more tokens, leading to increased latency and resource inefficiency. While existing strategies attempt to address this at the training stage, they often fail to generalize during inference.

### **Our Contributions**

1.  **Novel Load Balancing Strategy**:

    -   Introduced **Replication and Quantization (R&Q)** to alleviate load imbalance and maintain model accuracy:
        -   **Replicate Heavy-Hitter Experts**: Tokens are dispatched to any of the heavy-hitter experts and their replicas.
        -   **Quantization**: Heavy-hitter replicas are quantized to half precision, and less important experts are also quantized, maintaining memory constraints.
2.  **Load Imbalance Score**:

    -   Proposed a novel metric to evaluate load distribution across experts during inference, addressing a critical gap in prior research.
3.  **Empirical Results on Multiple Models**:
    -   Demonstrated consistent improvements in inference efficiency across various models, including Switch Transformers, LlaMa MoE and DeepSeek MoE.
    -   Provided comprehensive studies on models with varying sparsity configurations and routing strategies.


----------

## **Reviewers’ Feedback**

### **Positive Feedback**
1.  **Comprehensive experiments**:
    -   Reviewer Dy55:  "The author conducted experiments on various tasks and model types, creating a comprehensive overview of the impact of the proposed strategy on the performance of the model."
    -   Reviewer 5vEp:  "The authors have tested on various MoE base models."
     -  Reviewer CGsx:   "Near-original accuracy (at least on classification / MCQ tasks)"
2.  **Novel Insights**:
    -   Reviewer CGsx:  "I like how they distinguished between heavy-hitter experts and important experts, which could easily be confused as the same. They also conducted experiments to show that these concepts are distinct, although there is some correlation between them.."
    -  Reviewer UpDR: "The "Replicate and Quantize" strategy is a novel approach that dynamically addresses load imbalance in SMoE models without requiring extensive retraining."
     -  Reviewer Dy55:   "This paper studies an interesting problem, which is the load imbalance of MoE LLMs in inference scenarios."
3.  **Easy to implement**:
	-   Reviewer UpDR:  "The proposed strategy is plug-and-play, making it easy to integrate with existing models and practical for real-world applications."
	-   Reviewer CGsx:  "No retraining is required."
	 -   Reviewer 5vEp:   "The proposed idea is simple and sound." "Overall, this work is well-organized and easy to follow."
### **Remaining Concerns**

1.  **Scalability**:
    -   Reviewer CGsx: "Weak empirical analysis on more challenging tasks"
    -   Reviewer UpDR: "How does your method perform under different levels of model sparsity and varying numbers of experts in the SMoE models?"
    -   Reviewer 5vEp: "The scalability of the proposed method is encouraged to be evaluated or discussed."
2.  **Inference Efficiency**:
    -   Reviewer CGsx: "Requested further clarification on how R&Q affects inference speed and memory usage."
    -  Reviewer Dy55:  "The most important metrics, which are the inference performance of the proposed system (memory consumption, latency, hardware utilization, and so on), are not studied in this work. These are the most important reasons one would like to reduce the load imbalance."
3. **Detailed Ablation Study**:
	-   Reviewer UpDR:  "Could you conduct additional ablation studies to demonstrate the individual contributions of the replication and quantization components in your proposed method?"
	-  Reviewer 5vEp:   "an appropriate ablation study should be conducted to verify the effectiveness of both strategies on the most heavy experts and less important experts"

----------

---

> ### Author Response · Authors · 2024-12-04
>
> ## **Our Response to Concerns**
>
> ### **On Scalability**:
>
> -   **1. Add more challenging task**:
>
> 	TruthfulQA generation results and other challenging Tasks:
>
> | Model            | Type | bleu_acc | bleu_diff | rougeL_acc |
> | ---------------- | ---- | -------- | --------- | ---------- |
> | **DeepSeek MoE** | RQ   | 0.3133   | -7.7523   | 0.2876     |
> |                  | Raw  | 0.3121   | -7.7476   | 0.2901     |
> | **Llama MoE**    | RQ   | 0.2925   | -8.1597   | 0.2546     |
> |                  | Raw  | 0.2913   | -8.5101   | 0.2436     |
>
> | Dataset                | Type | Value                       | Type | Metric            | Value   |
> |------------------------|------|-----------------------------|------|-------------------|---------|
> | **codexglue_code2text**| Raw  | 2.6119107948506235          | Raw  | smoothed_bleu_4   | 1.5517  |
> |                        | RQ   | 1.9426032317889572          |  RQ  |                   | 1.5463  |
> | **coqa**               | Raw  | 2.0412320534522754          | Raw   | f1                | 0.0104  |
> |                        | RQ   | 1.6487805751851754          |RQ    |                   | 0.0106  |
> | **wikitext**           | Raw  | 2.090017361111111           | Raw   | byte_perplexity   | 17.2096 |
> |                        | RQ   | 1.6235243055555557          |  RQ    |                   | 17.2826 |
> |                        |      |                             | Raw   | bits_per_byte     | 4.1051  |
> |                        |      |                             |RQ|                   | 4.1112  |
>
> - **2. Clarify the previous experiments setting**
> 	We have conducted experiment in the paper with the following models with different sparsity.
> 	**Switch Transformer 8 experts (Sparsity 1/8):**
> 	total have 8 experts in each layer, only choose 1 expert for each token
> 	**Switch Transformer 16 experts  (Sparsity 1/16):**
> 	total have 16 experts in each layer, only choose 1 expert for each token.
> 	**Llama MoE 8 experts (Sparsity 2/8):**
> 	total have 8 experts in each layer, choose 2 experts for each token
> 	**Deepseek MoE (Sparsity 1+3/63):**
> 	total has 64 experts in each layer: 63 isolated experts and 1 shared expert.  In each routing, the system will select three experts from the 63 isolated experts, always using the shared expert. Therefore, our strategy is exclusively applied to the 63 isolated experts.
> - **3. Scalability in streaming data**
>   In Figure 3, Panels (a) and (b) illustrate the load balance scores across multiple timesteps under two distinct routing strategies. Panel (a) demonstrates the case where the system utilizes cumulative information aggregated from all prior timesteps, providing a holistic approach to decision-making. Conversely, Panel (b) focuses on a scenario where only information from the immediately preceding timestep is leveraged, showcasing a more localized decision-making approach.
>
>     The purpose of this setup is to identify the most important "heavy-hitter" experts from the less important ones. These decisions depend directly on the input data at each timestep. To mimic how data arrives in a real-world streaming scenario, we divided the MMLU dataset into 10 parts and set the timestep count to 10. This approach allows us to simulate the flow of data over time and study how it affects the routing strategy in a manageable and realistic way.
>
> ### **On Inference Efficiency**:
> Our method ensures memory usage does not increase even when duplicating new experts. After duplication, we quantize the new experts and further quantize less-important ones to half precision.
>
> Regarding inference speed, in each MoE layer, the inference time is determined by the slowest expert, which handles the largest number of tokens. Below, we provide an example to illustrate this behavior.
>
> 	### Example for Each MoE Layer
>
> 	Premise:
>
> 	- Number of Experts: 4
> 	- Number of Tokens: 80
> 	- Processing Capacity per Expert: Each expert processes (80 / 4 = 20) samples per unit time (T).
>
> 	---
>
> 	Original Routing Results:
>
> 	- Experts' Token Allocation: [18, 30, 25, 7]
> 	- Inference Time (Slowest Expert):
> 	  Expert 1 handles 30 tokens:
> 	  30 / 20 = 1.5 T
>
> 	---
>
> 	After Rerouting & Quantization (R & Q):
>
> 	- Experts' Token Allocation: [18, [17, 13], 25, 7]
> 	- Inference Time (Slowest Expert):
> 	  Expert 2 handles 25 tokens:
> 	  25 / 20 = 1.25 T
>
> 	Result:
> 	By applying our method, the inference time is reduced, resulting in a speed-up of 0.25 T.
>
> ---

---

> > ### Author Response · Authors · 2024-12-04
> >
> > ### **On Ablation Study**:
> >
> > | Model & Configuration            | Hellaswag            | MMLU                | PIQA                | Truthful QA         | Winogrande         |
> > |----------------------------------|----------------------|---------------------|---------------------|---------------------|---------------------|
> > | **Switch 8**                     |                      |                     |                     |                     |                     |
> > | Only Quant Less-Important Experts | 0.2795 ± 0.0045     | 0.2295 ± 0.0035     | 0.5751 ± 0.0115     | 0.3605 ± 0.0110     | 0.5138 ± 0.014      |
> > | Replicate and Quant replicate one | 0.2749 ± 0.0045     | 0.2498 ± 0.0037     | 0.5811 ± 0.0115     | 0.3635 ± 0.0110     | 0.4917 ± 0.0141     |
> > | Quant All Experts                 | 0.2641 ± 0.0044     | 0.2295 ± 0.0035     | 0.5490 ± 0.0116     | 0.3775 ± 0.0112     | 0.5185 ± 0.014      |
> > | R&D                               | 0.2763 ± 0.0045     | 0.2522 ± 0.0037     | 0.5832 ± 0.0115     | 0.3640 ± 0.0110     | 0.4917 ± 0.0141     |
> > | Quant All Model                   | 0.2927 ± 0.0045     | 0.2295 ± 0.0035     | 0.5952 ± 0.0115     | 0.3706 ± 0.0111     | 0.5091 ± 0.0141     |
> > | **Switch 16**                     |                      |                     |                     |                     |                     |
> > | Only Quant Less-Important Experts | 0.2820 ± 0.0045     | 0.2295 ± 0.0035     | 0.5577 ± 0.0116     | 0.3914 ± 0.0114     | 0.4964 ± 0.0141     |
> > | Replicate and Quant replicate one | 0.2864 ± 0.0045     | 0.2490 ± 0.0036     | 0.5490 ± 0.0116     | 0.3669 ± 0.0112     | 0.483 ± 0.014       |
> > | Quant All Experts                 | 0.2595 ± 0.0044     | 0.2295 ± 0.0035     | 0.5582 ± 0.0116     | 0.3721 ± 0.0110     | 0.5091 ± 0.0141     |
> > | R&D                               | 0.2872 ± 0.0045     | 0.2495 ± 0.0036     | 0.5501 ± 0.0116     | 0.3694 ± 0.0112     | 0.4854 ± 0.014      |
> > | Quant All Model                   | 0.2768 ± 0.0045     | 0.2295 ± 0.0035     | 0.5539 ± 0.0116     | 0.3726 ± 0.0112     | 0.4901 ± 0.014      |
> >
> > ----------
> >
> > ### **Why This Work Matters**
> >
> > 1.  **Practical Load Balancing Strategy for Inference**:
> >     -  Significantly reduces inference time while maintaining accuracy and preserving memory usage.
> >
> > 2.  **Foundational for Future Research**:
> >     -   Introduces the novel _Load Imbalance Score_ metric, providing a valuable tool for future studies on SMoE model efficiency.
> >
> > 3.  **Adaptable and Scalable**:
> >     -  Demonstrates practical applicability across diverse models, tasks, and real-world streaming data scenarios.
> >
> >
> > ----------
> >
> > ### **Conclusion**
> >
> > We hope the above overview may provide our AC and reviewers with a concise way to navigate through the mass information present on this page.
> > Further, we sincerely hope our appreciation of simple but effective design, as well as our novel observations and insights into load imbalance in SMoE model, can be shared with you and our fellow scholars in this long-overlooked but important field of SMoE model.
> >
> > Sincerely,
> >
> > Paper Authors

---

### Meta-Review · Area_Chair_j2x2 · 2024-12-10

**Metareview:**

The submission presents a load-balancing strategy for mixture of experts models involving quantization of experts.  The reviewers indicated that the submission was borderline or below the acceptance threshold, with a majority indicating that the submission should be rejected.  Reviewer Dy55 indicates that the empirical evaluation is incomplete.  Reviewer CGsx indicates that the connection between the design choices and speedups is inadequately explained.  Reviewer 5vEp was appreciative of the additional explanations, but still felt that the contribution was not significant enough in its current form to recommend acceptance.

**Additional Comments On Reviewer Discussion:**

The reviewer discussion was active on all parts, with substantial responses from the authors, which were appreciated by the reviewers.  The reviewers have suggested these additional explanations be included in future revisions of the work should it be submitted to another conference in the future.

---

### Decision · Program_Chairs · 2025-01-22

Reject